



# Intraseasonal summer rainfall variability over China in the MetUM GA6 and GC2 configurations

Claudia Christine Stephan[1], Nicholas P Klingaman[1], Pier Luigi Vidale[1], Andrew G Turner[1,2], Marie-Estelle Demory[1,3], and Liang Guo[1]

[1]National Centre for Atmospheric Science – Climate, Department of Meteorology, University of Reading, P.O. Box 243, Reading RG6 6BB, United Kingdom
[2]Department of Meteorology, University of Reading, P.O. Box 243, Reading RG6 6BB, United Kingdom
[3]Center for Space and Habitability, University of Bern, Gesellschaftsstrasse 6, 3012 Bern, Switzerland

*Correspondence to:* Claudia Stephan (c.c.stephan@reading.ac.uk)

**Abstract.** The simulation of intraseasonal precipitation variability over China in extended summer (May–October) is evaluated based on six climate simulations of the Met Office Unified Model. Two simulations use the Global Atmosphere 6.0 (GA6), and four the Global Coupled 2.0 (GC2) configuration. Model biases are large, such that mean precipitation and intraseasonal variability reach twice their observed values, particularly in southern China. To test the impact of air-sea coupling and horizontal

5 resolution, GA6 and GC2 at horizontal resolutions corresponding to ∼25, 60 and 135 km at 50° N are analyzed. Increasing the horizontal resolution and adding air-sea coupling have little effect on these biases. Pre-monsoon rainfall in the Yangtze River basin is too strong in all simulations. Simulated rainfall amounts in June are too high along the southern coast and persist in the coastal region through July, with only a weak northward progression. The observed northward propagation of the Meiyu/Baiu/Changma rainband from spring to late summer is poor in all GA6 and GC2 simulations. To assess how well the

10 MetUM simulates spatial patterns of temporally coherent precipitation, empirical orthogonal teleconnection (EOT) analysis is applied to pentad-mean precipitation. Patterns are connected to large-scale processes by regressing atmospheric fields onto the EOT pentad timeseries. Most observed patterns of intraseasonal rainfall variability are found in all simulations, including the associated observed mechanisms. This suggests that GA6 and GC2 may provide useful predictions of summer intraseasonal variability, despite their substantial biases in mean precipitation and overall intraseasonal variance.

## 1 Introduction

About half of the annual China-wide precipitation falls between June and August. Strong summer precipitation events can cause severe flooding and disrupt the nation's economy (Huang et al., 1998). The summer monsoon over the East Asian northwest Pacific sector is modulated by active-break cycles (Chen and Murakami, 1988; Sumathipala and Murakami, 2010). These active-break cycles are themselves modulated by organized dynamically-coupled convective systems in the tropical and sub-

20 tropical Indian Ocean and western tropical Pacific. These convective systems mainly propagate eastward along the equator, but also northward to the subtropics (Kayano and Kousky, 1999; Hsu and Weng, 2001). Because of their oscillatory nature with typical periods of 10–90 days they have also been termed the boreal summer intraseasonal oscillation (BSISO; Sect. 2.3). The



BSISO is the summer season equivalent of a recurring convective system that appears mainly in boreal winter and propagates eastward from the tropical Indian Ocean into the Pacific, the Madden-Julian Oscillation (MJO; Madden and Julian, 1972). BSISO activity affects the monsoon onset and withdrawal, and the intensity of precipitation along the East Asian monsoon front (Hsu, 2005; Ding, 2007).

In addition to the large variety of physical mechanisms, the complicated topography of East Asia is a challenge for the simulation of precipitation. Despite efforts to improve General Circulation Models (GCMs) by adding complexity to the representation of physical processes, little improvements are seen between the third and fifth Coupled Model Intercomparison Projects (CMIP3 and CMIP5, Meehl et al., 2007; Taylor et al., 2012) in terms of the simulation of the mean state over East Asia (Sperber et al., 2013). Common mean state biases in contemporary coupled GCMs include cold temperatures and ex-

cessive precipitation (Jiang et al., 2005), and an underestimated southeast-northwest precipitation gradient across East Asia (Jiang et al., 2016). In southern China CMIP5 models overestimate precipitation in both summer and winter, including the intensity of extreme precipitation events (Liu et al., 2014). This suggests that GCMs struggle to capture monsoonal and convective events (IPCC, 2007).

There is no consensus on the effects of finer atmospheric horizontal resolution. Using targeted regional numerical exper-

iments, Gao et al. (2006) found evidence that the simulation of the distribution of precipitation over China in the National Center for Atmospheric Research's (NCAR) Regional Climate Model version 2 (RegCM2) improved at horizontal resolutions smaller than 60 km, testing 45–360 km. They were able to attribute this improvement to the better simulation of associated mechanisms. The Community Atmosphere Model version 5 at truncations of T42 (200 km at 50° N), T106 (80 km), and T266 (50 km) showed reduced rainfall biases near mountains due to more realistic orography (Li et al., 2015). In contrast,

Song and Zhou (2014a) found summer precipitation associated with the East Asian summer monsoon (EASM) in atmosphere-only CMIP3 and CMIP5 simulations to be insensitive to horizontal resolution (∼20–500 km). Jiang et al. (2016), analyzing CMIP3–CMIP5 simulations at horizontal resolutions of ∼60–620 km, and Chen and Frauenfeld (2014), analyzing CMIP5 simulations at horizontal resolutions of ∼210–830 km, obtained a similar result for coupled GCMs. Instead, Song and Zhou (2014a) found varying performance for different atmosphere-only CMIP3 and CMIP5 models at the same resolution (varying

from ∼220–450 km); based on this, they suggested that the model formulation may play a more important role than resolution. In MetUM GA3 summer rainfall biases inside the Asian monsoon domain 50–180° E, 20° S–40° N (Johnson et al., 2016), and the East Asian monsoon domain 120–180° E, 0–40° N (Ogata et al., 2017), increase slightly when changing resolution from N96 (135 km at 50° N) to N216 (60 km), and N216 to N512 (25 km).

In comparing atmosphere-only to coupled GCM simulations over the summer monsoon regions, autocorrelations of day-

to-day rainfall were seen to improve with coupling (Misra, 2008). Air–sea coupling was also shown to significantly improve the simulation of the BSISO (Kemball-Cook et al., 2002; Fu et al., 2003; Fu and Wang, 2004a; Klingaman et al., 2011; DeMott et al., 2014). MetUM GC2 simulations at resolutions of 135 and 25 km showed improved mean sea-surface temperature and low-level specific humidity with finer atmospheric resolution (Fang et al., 2017). Fang et al. (2017) also reported that the model simulated well the characteristics of the BSISO, including its dominant spatial patterns, cyclical evolution and north-

ward propagation. They reported further improvements in the northward propagation of precipitation with periods of 20–70



days with finer resolution due to better resolved air–sea interactions. However, the simulation of mean precipitation did not improve with finer atmospheric resolution.

It is difficult to isolate the direct effects of finer horizontal resolution and the addition of air-sea coupling when the model formulation is also changed, as is the case in the CMIP experiments. Therefore, we use the same six MetUM coupled and atmosphere-only simulations at 135–25 km resolution that were analyzed for interannual variability of East Asian precipitation (Stephan et al., 2017c). For the first time, GCM simulations over China are assessed not only for overall summer intraseasonal variability, but also in terms of being able to reproduce leading patterns of temporally coherent precipitation variability, which are defined by empirical orthogonal teleconnection (EOT) analysis. In their analysis of 1982–2007 pentad precipitation, Stephan et al. (2017b) performed EOT analysis to determine the mechanisms responsible for intraseasonal rainfall variability in China. The leading three observed patterns (Obs-1, Obs-2, Obs-3) explain 14 % of the total space-time variance and are located in southern China, central eastern China (with variability of opposite phase along the southeast coast), and the southeast coast. The total explained variance is only 14 % because of the regional nature of intraseasonal summer precipitation systems. The leading patterns were all associated with BSISO variability (Sect. 5). We use results from Stephan et al. (2017b) as the basis to assess the MetUM simulations. Due to its spatial coherence such variability is particularly important for understanding the risks of droughts and flooding at the regional scale. Also based on Stephan et al. (2017b), Stephan et al. (2018) evaluated the GA6 and GC2 models used here for their ability to simulate winter intraseasonal variability of precipitation over China. GA6 and GC2 simulate well regional variability in extended winter, including associated northern-hemisphere wave dynamics.

Our MetUM GA6 and GC2 model simulations, observation data and EOT analysis are described in Sect. 2. Section 3 discusses biases in the mean seasonal cycle in these simulations. Section 4 examines subseasonal variability on different timescales. The simulation of coherent precipitation patterns derived from EOT analysis is discussed in Sect. 5. A discussion and summary are presented in Sect. 6.

## 2 Data and methods

### 2.1 MetUM simulations

We analyze six MetUM simulations, two AMIP-style simulations (A96, A216) from 1982–2008 and four 100-year coupled simulations (C96, C216, C512a and C512b). In this naming convection 'A' or 'C' refer to atmosphere-only or coupled, followed by the nodal number that indicates the atmospheric horizontal resolution (N96: $1.875° \times 1.25°$, 135 km at 50° N, N216: $0.83° \times 0.55°$, 60 km, N512: $0.35° \times 0.23°$, 25 km). The ocean uses the ORCA025 tri-polar grid (Madec, 2008) with a fixed resolution of one-quarter degree, 75 vertical levels with a 1 m top level, and a coupling frequency of 3 hours. Table 1 summarizes key information about the simulations.

Atmosphere-only runs use the MetUM Global Atmosphere 6.0 configuration (GA6; Walters et al., 2017), with 1982–2008 observed solar, greenhouse gas and aerosol forcings; their monthly mean sea surface temperatures are based on the Reynolds product (Reynolds et al., 2007). Coupled simulations are performed with the MetUM Global Coupled configuration 2.0 (GC2; Williams et al., 2015). The initialization of C96 and C216 uses a spin-up simulation for sea-ice and land surface conditions,



and ocean conditions from a present-day data set (EN3; Ingleby and Huddleston, 2007). Initial ocean conditions for C512a and C512b are taken from a different coupled N512 MetUM simulation. To initialize in different phases of decadal variability, the initial conditions for C512a and C512b are offset by 55 years. The four GC2 simulations use constant 1990 greenhouse gas concentrations, aerosols, emissions and solar forcing.

## 2.2 Observational and reanalysis data

Daily precipitation over China is obtained from the Asian Precipitation - Highly-Resolved Observational Data Integration Towards Evaluation (APHRODITE; Yatagai et al., 2012) data set, with a resolution of $0.5° \times 0.5°$. Please refer to Stephan et al. (2017a) for a more detailed description of this data set. For consistency with Stephan et al. (2017b) and with the GA6 AMIP experiments, we use data for 26 years of extended summer (MJJASO), covering May 1982 – October 2007. We focus on the extended summer season because the patterns of coherent precipitation variability identified in Stephan et al. (2017a) are connected to the BSISO, which is defined for MJJASO (Lee et al., 2013).

We use horizontal wind at 850 hPa and geopotential height at 500 hPa ($Z_{500}$) from the European Centre for Medium-Range Weather Forecasts Interim global reanalysis (ERA-Interim; Dee et al., 2011) from 1982–2007 at $0.7° \times 0.7°$ resolution. Pentad means of $2.5° \times 2.5°$ interpolated satellite-retrieved outgoing longwave radiation data (OLR; Liebmann and Smith, 1996) for 1982-2007 are used as a measure of convective activity.

## 2.3 BSISO indices

As indicators of the extended summer intraseasonal oscillation (BSISO) we use the two sets of BSISO indices, BSISO-1 and BSISO-2, defined by Lee et al. (2013). They are derived from multivariate empirical orthogonal function (MV-EOF) analysis of normalized daily mean OLR and 850 hPa zonal wind (NCEP Reanalysis II; Kanamitsu et al., 2002) anomalies over the Asian summer monsoon region (10° S-40° N, 40° E-160° E) based on May to October data for the period 1981–2010. To compute anomalies, Lee et al. (2013) first remove the mean of the annual cycle, the average of the leading three harmonics of the annual cycle, and then the running mean of the previous 120 days. The normalized principal component timeseries of the leading two MV-EOFs (PC1,PC2) define BSISO-1, and the third and fourth (PC3,PC4) define BSISO-2. BSISO-1 and BSISO-2 amplitudes are expressed as $(PC1^2 + PC2^2)^{1/2}$ and $(PC3^2 + PC4^2)^{1/2}$, respectively. Composites of OLR and 850 hPa wind for each phase of BSISO-1 and BSISO-2 are shown in Figs 9 and 10 in Lee et al. (2013).

We use pentad-mean BSISO indices computed from daily values. To compute BSISO indices from model output, we compute anomalies of simulated fields by performing the same steps as described above. The anomaly fields are normalized as in the MV-EOF analysis and then projected onto the EOFs of Lee et al. (2013). The resulting pseudo-principal component timeseries are then used to compute pentad-mean BSISO indices, as we do for observations. We project model data onto observed EOFs to test how well the simulations represent observed intraseasonal variability. This approach more fairly compares the simulations, as the EOF structures vary by simulation and between model and observations (Klingaman and Woolnough, 2014).



## 2.4 Empirical Orthogonal Teleconnections

Empirical orthogonal teleconnections (EOTs; Van den Dool et al., 2000) are used to find regions of temporally coherent intraseasonal precipitation variability (IPV). The method identifies points (base points) that explain the most variance in the rainfall timeseries averaged over the domain and returns timeseries that are mutually orthogonal (Smith, 2004). Hence, the

base points are located in particular places in China. Associated local and large-scale mechanisms are found by regressing atmospheric fields onto the corresponding EOT timeseries. EOT analysis has successfully been applied to China (Stephan et al., 2017a, b) and other parts of the globe (Smith, 2004; Rotstayn et al., 2010; Klingaman et al., 2013; King et al., 2014). The EOT algorithm, which is described in detail in Stephan et al. (2017a, b), is applied to observed and simulated pentad-rainfall anomalies interpolated to the APHRODITE grid, so that the observed EOT patterns remain identical to the ones identified in

Stephan et al. (2017b). To test the significance in regressions of atmospheric fields against normalized EOT timeseries we use Spearman's rank correlations to account for the non-Gaussian distribution of rainfall data.

## 2.5 Precipitation spectra

To examine daily and pentad precipitation intensities, we compute the contributions of discrete bins of precipitation intensity to daily and 5-day precipitation totals, respectively. We use the exponential bins defined in Klingaman et al. (2017). Observed and

simulated data are first averaged onto a common $2° \times 2°$ grid inside southern China (22–32° N, 103–119° E). We compute the spectrum using precipitation data at all grid points inside the domain. By definition, the sum across all bins is unity. Therefore, the spectrum shows the relative importance of precipitation events in a given intensity bin to the total precipitation, but does not contain information about the frequency of precipitation. Furthermore, as a measure of the typical duration of precipitation events, we compute the autocorrelation function of daily precipitation at each grid point inside the southern China domain and

then average.

## 3 Mean state biases

In terms of absolute values it is expected that rainfall variance is large where mean rainfall is large. Therefore, we begin our model assessment with a discussion of mean state biases and biases in the simulation of subseasonal features of the mean seasonal cycle.

Substantial biases in summer mean precipitation and IPV are present in MetUM GA6 and GC2 (Fig. 1). Observed mean rainfall and IPV decrease from southeast to northwest China. In all simulations isolines of mean rainfall are too zonal and mean rainfall is as much as twice the observed amount in large parts of China. These biases are hardly affected by resolution or coupling, consistent with previous studies (Song and Zhou, 2014a; Jiang et al., 2016; Johnson et al., 2016; Ogata et al., 2017). The spatial pattern of IPV matches observations better than mean precipitation. All simulations produce a southeast–northwest

gradient of IPV, albeit with overestimated IPV in southeast and southwest China. IPV improves (reduces) with coupling, but does not systematically improve with finer resolution.





To understand the origins of biases in mean precipitation we next examine monthly mean wind fields (Fig. 2) and the seasonal cycle of precipitation, illustrating March–September rainfall over 105–120° E using Hovmöller diagrams of pentad rainfall (Fig. 3). Fig. 3 shows March–September to illustrate the full cycle of the EASM. Positive precipitation biases along the Yangtze River valley develop in April–May. They are related to an anticyclonic circulation bias over the Philippines with

strong southerlies over the South China Sea (Fig. 2a–c).

As in observations, westerlies develop over South Asia in June and strengthen in July and August (Fig. 2d–l). In simulations, particularly in GC2, these extend too far eastward. These biases are linked with a poor simulation of the monsoon rainband. In observations, the precipitation frontal zone shows a seasonal march that is connected to the northward advancing Pacific anticyclone (Ninomiya and Muraki, 1986; Kato, 1989). Observed rainfall shows a smooth northward propagation from spring

to late summer. The 5 mm day$^{-1}$ contour progresses from 27° N in March to 33° N at the end of July. The observed peak rainfall occurs between 25° N–35° N in June–July (Fig. 3). Simulated rainfall in June is too strong along the southern coast and persists in the coastal region (20° N-25° N) through July, with only a weak northward progression during July, as is common in others models (Boo et al., 2011).

In September and October wind and precipitation biases become small (Fig. 2m–r).

## 15   4   IPV on different timescales

Figure 1 shows that biases in simulated IPV are maximum in southern China and along the Yangtze River valley. The analysis of mean rainfall in the previous section suggests that IPV biases are associated with excessive total precipitation. We now investigate what timescales are associated with IPV biases.

To better compare IPV, Fig. 4 shows the meridional profiles of bandpass-filtered precipitation variance averaged between

103–119° E. To match the typical period of BSISO-1 and BSISO-2, we filter for 30–80 and 10–30 day variability, respectively. The shape of the meridional variance profile is correctly simulated, but magnitudes are 2–3 times larger than observed, especially in southern China (Fig. 4a,b). Coupling reduces these biases, as was also seen for total precipitation (Fig. 1).

Simulations overestimate daily and 5-day rainfall amounts (Fig. 4c). Overestimated 5-day rainfall totals partly result from a lack of day-to-day variability; the decorrelation time of daily rainfall averaged over all points in southern China (22–32° N,

103–119° E) is larger in the simulations than in observations (Fig. 4d).

To further investigate IPV with periods of 30–80 days, Figs 5 and 6 show the regressed space–time evolution of precipitation and 850 hPa wind with periods of 30–80 days. Observations show that a coupled circulation-precipitation system propagates north- and northwestward from the South China Sea and the western North Pacific. Peak precipitation over the South China Sea is associated with a strong anomalous cyclone over southeast Asia. Divergent anomalous easterly and northeasterly winds

on the northern flank of this cyclone are associated with reduced precipitation over the Yangtze basin. At lead-lag times of ±4 pentads, respectively, this pattern reverses.

Simulations generally capture the observed evolution but produce a southwest to northeast tilt. This creates wet anomalies at ±4 pentads, respectively, and dry anomalies at pentad 0 in southern China, where observed anomalies are close to zero. A





similar tilt of the BSISO was noted by Fang et al. (2017) for GA3 at N96 and N216 who examined 20–70 day filtered OLR and 850 hPa wind regressed against rainfall in 10—22° N, 110–130° E, i.e. in a box that extends 10° further east than in this study. Magnitudes of precipitation and wind anomalies at ±6 pentads are overestimated in GA6 but are more realistic in GC2. However, at ±4 pentads and lag zero GA6 and GC2 both overestimate precipitation anomalies.

## 5  Coherent patterns of IPV

In the previous sections we have established that mean-precipitation biases in MetUM GA6 and GC2 are substantial, and that these are associated with circulation biases. Averaged over extended summer, GA6 and GC2 produce too heavy daily rainfall. These biases are mostly insensitive to atmospheric horizontal resolution and air-sea coupling. As expected, IPV is large where mean precipitation is large. But, unlike mean precipitation, IPV improves slightly with coupling. We now discuss the fidelity of the simulation of leading spatial patterns of coherent IPV reported by Stephan et al. (2017b), derived from EOT analysis. We test whether GA6 and GC2 can produce observed patterns despite their biases in mean precipitation, overall IPV, and the seasonal cycle. Furthermore, we investigate whether the simulations capture observed physical mechanisms associated with the leading patterns of coherent IPV (i.e. the right variability for the right reasons).

Simulated precipitation is interpolated to the APHRODITE grid and regressed against simulated EOT timeseries to test how well the simulations capture observed patterns. We consider two patterns a match when the linear Pearson pattern correlation coefficient between simulated and observed regression maps exceeds 0.62. This threshold is chosen because EOTs with smaller pattern correlations all peak in areas that are far away from the observed patterns. All matching patterns are shown in Fig. 7. To summarize the model performance at simulating EOTs, Table 2 lists the explained variance, the pattern correlations, and the standard deviations of the timeseries of each simulated and observed EOT.

All simulations produce Obs-1 with correlation coefficients exceeding 0.74. In A96 the pattern is found in the second EOT. Greater variability in the simulated, relative to the observed timeseries, is consistent with the large positive biases in IPV in south China (Fig. 1). Simulations produce Obs-2 as their second or third EOT. Pattern correlations are higher in the atmosphere-only simulations (0.77, A96; 0.90, A216), because the pattern is shifted southeastward in C96 and C216 and northward in C512a and C512b. Obs-3 is only found in A216.

We now compare simulated and observed atmospheric anomalies associated with each EOT. Obs-1 is associated with suppressed convection over the South China Sea, the Philippines and the northwest Pacific (Fig. 8a). There are easterly low-level wind anomalies across the Philippines and southwesterly onshore winds into the region of southeast China that experiences higher precipitation. Similar OLR and circulation anomalies are found in all simulations (Fig. 8b-g).

Obs-3 is similar to Obs-1 (Fig. 8h), but in Obs-3 there is a strong anomalous cyclonic circulation over southeast China and the southwesterly onshore winds are situated further east along the coast, consistent with the peak of Obs-3 rainfall being located east of the rainfall of Obs-1. A216 is able to produce these wind anomalies (Fig. 8i).

Obs-2 is associated with suppressed convection in the northwest Pacific (Fig. 9). It is the only pattern with significant extra-tropical $Z_{500}$ anomalies. In the simultaneous pentad, anomalously high $Z_{500}$ is present over Russia, low $Z_{500}$ over Mongolia





and northern China and high $Z_{500}$ over southeast China and the northwest Pacific. The positive OLR anomaly is already present in the two preceding pentads, but is located further east in the northwest Pacific (not shown). Extratropical perturbations associated with Obs-2 appear one pentad in advance and are characterized by high $Z_{500}$ in western Russia, low $Z_{500}$ in northwest China and Mongolia and high $Z_{500}$ over the northwest Pacific. A96, C96 and C216 show similar OLR and $Z_{500}$ anomalies in

the simultaneous and preceding pentad. In A216 and C512 the positive OLR anomaly is not present. In addition, C512 show stronger extratropical perturbations with strong signals at a lead time of 2 pentads (not shown).

The percentages shown by the histograms in Fig. 10 indicate the fractions of the wettest decile of pentads that occur during phases 1–5 of BSISO-1 (top) and phases 4–8 of BSISO-2 (bottom), when these are statistically significantly increased at the 5 % level compared to climatology, as indicated by a two-sided binomial test. Wet Obs-1 events occur preferentially in BSISO-1

phase 4 and BSISO-2 phase 5, those of Obs-2 in BSISO-1 phase 3, and those of Obs-3 in BSISO-2 phase 4.

The similarity of A96, C96 and C216 with observed EOT-2 is reflected in their BSISO phase preference of BSISO-1 phase 3, as in observations. In addition, C216, C512a and C512b have a slight preference for BSISO-2 phase 2, as observed EOT-1. Except for these matches, there exists no further agreement in BSISO phase preferences between simulations and observations, despite the good match of the simulated and observed rainfall patterns. This is due to substantial biases in the precipitation

patterns associated with specific BSISO phases. Figure 11 shows a representative example of these biases for BSISO-1 phase 3. Even though Figs 5 and 6 show a relatively good simulation of north and northwestward propagating BSISO anomalies, this does not translate into precipitation anomalies associated with specific BSISO phases.

Nevertheless, MetUM GA6 and GC2 capture coherent patterns of observed precipitation variability and the associated mechanisms. This shows that the model simulates well the tropical and extratropical perturbations associated with regional

rainfall.

## 6  Discussion and summary

We analyzed two MetUM GA6 simulations at resolutions of N96 and N216 (referred to as A96 and A216, respectively) and four MetUM GA6 simulations at resolutions of N96–N512 (C96, C216, C512a and C512b, Table 1) in terms of their fidelity at simulating mean rainfall and intraseasonal precipitation variability (IPV) in extended summer (May-Oct). GCM simulations

over China were also assessed for their ability to produce the leading patterns of IPV, defined by empirical orthogonal teleconnections (EOTs), and associated physical processes. Simulating such coherent regional variability is important if models are to be used for assessing the risks of flooding at the regional scale.

The fact that the two C512 simulations are very similar suggests that the model internal variability is small, although these are only two simulations at one particular resolution. Therefore, it is plausible that the differences between simulations are

indeed due to changes in resolution and air-sea coupling.

Mean precipitation and IPV tend to be twice the observed values in the MetUM (Fig. 1), consistent with other models (Jiang et al., 2005; Liu et al., 2014; Jiang et al., 2016). Positive biases in southeast China are related to too heavy daily rainfall rates. In late spring an anticyclonic circulation bias over the Philippines with strong southerlies over the South China Sea



contributes to wet biases in southeast China. This anticyclonic circulation bias is strongest in April and May. It is associated with wet biases along the Yangtze valley in March–May; pre-monsoon rainfall in the Yangtze River basin is too strong.

A second bias in the MetUM is a zonal extension of the South Asian summer monsoon low level westerly jet. Related to this, the observed northward propagation of the monsoon front from spring to late summer is poorly simulated regardless of resolution or coupling. Simulated rainfall in June is too strong along the southern coast and persists in the coastal region through July, with only a weak northward progression during July.

Spin-up experiments with the Met Office's seasonal forecasting system (GloSea5; MacLachlan et al., 2015) indicate that the spring anticyclonic circulation bias and the zonal extension of the South Asian summer monsoon low level westerly jet are both associated with excessive rainfall to the east of the Philippines, which develops immediately and is caused by the behavior of the convection scheme [personal communication with Dr. Gill Martin].

Air-sea coupling and resolution have little effect on mean precipitation and IPV. This finding is in agreement with Martin et al. (2017) who reported that tropical rainfall in MetUM GA6 is insensitive to horizontal resolution in boreal summer because all resolutions use parameterized convection. We found that adding coupling improves 10–30-day and 30–80-day variance.

Unlike for interannual variability (Stephan et al., 2017c), MetUM GA6 and GC2 are capable of simulating coherent regional patterns of observed IPV including the correct physical mechanisms. This implies that GA6 and GC2 have difficulty capturing large-scale teleconnections that are important for interannual variability, but can accurately simulate the physical mechanisms that are relevant at subseasonal timescales. Due to biases in the precipitation patterns associated with specific BSISO phases, simulated EOTs have different BSISO phase preferences than observed EOTs. EOT pattern correlation coefficients do not systematically improve with coupling or resolution (Table 2). It is remarkable that MetUM GA6 and GC2 simulate well regional variability despite substantial biases in mean precipitation. We can attribute some of the skill at simulating variability to the reasonable simulation of the oscillatory dynamic flow. Large biases in mean precipitation are very likely imposed by physical parameterizations, e.g. the use of a convective parameterization.

The above shows that model biases are seasonally and regionally dependent and vary with the timescales that are considered. The sensitivity of biases to resolution and air-sea coupling also depends on region, season, timescale, and on the formulation of unresolved physics. This explains why there exists no consensus on the benefits of higher resolution and the addition of air-sea coupling for the region and the timescales considered. To improve models it is necessary to identify the causes of biases in each model configuration separately. Other GCMs may show very different effects of coupling and resolution, particularly in the summer season, because of their different model formulation and physics parameterization schemes.

*Competing interests.* No competing interests are present.

*Code and data availability.* Data and code will be made available upon request through JASMIN (http://www.jasmin.ac.uk/).



*Acknowledgements.* This work and its contributors (Claudia Christine Stephan, Pier Luigi Vidale, Andrew Turner, Marie-Estelle Demory and Liang Guo) were supported by the UK-China Research & Innovation Partnership Fund through the Met Office Climate Science for Service Partnership (CSSP) China as part of the Newton Fund. Nicholas Klingaman was supported by an Independent Research Fellowship from the Natural Environment Research Council (NE/L010976/1). The high-resolution model C512 was developed by the JWCRP-HRCM group. The C512 simulations were supported by the NERC HPC grants FEBBRAIO and FEBBRAIO-2 (NE/R/H9/37), and they were performed on the UK National Supercomputing Service ARCHER by Prof. Pier Luigi Vidale and Karthee Sivalingam. APHRODITE data are available from http://www.chikyu.ac.jp/precip/. OLR data are provided by the NOAA/OAR/ESRL PSD, Boulder, Colorado, USA, at http://www.esrl.noaa.gov/psd/.



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



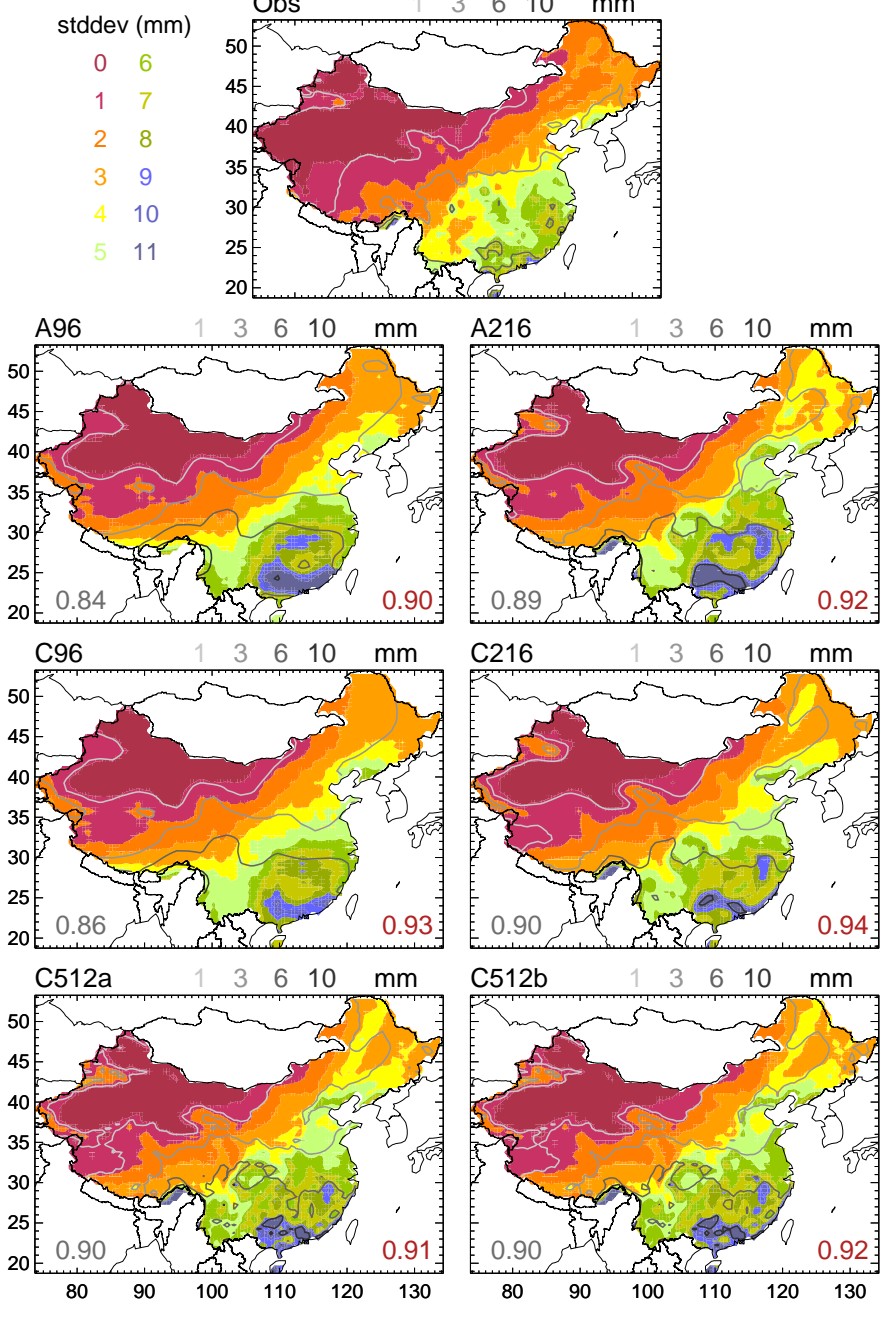

**Figure 1.** Climatological MJJASO total precipitation (gray contours) and standard deviation of pentad precipitation (shading) for 1982–2007 observations (top) and the full length of each simulation. Units are mm day$^{-1}$. All data are interpolated to the APHRODITE grid. The gray number on the bottom left is the linear pattern correlation coefficient between simulated and observed total precipitation, the red number on the bottom right is that for the standard deviation of pentad precipitation.



**Figure 2.** Climatological monthly mean 850 hPa wind for observations (left), A216 (middle) and C216 (right). Data were first interpolated to a $2° \times 2°$ grid. Shading shows the absolute horizontal wind velocity. Linear pattern correlation coefficients between simulated and observed zonal and meridional wind are denoted by ru and rv, respectively.





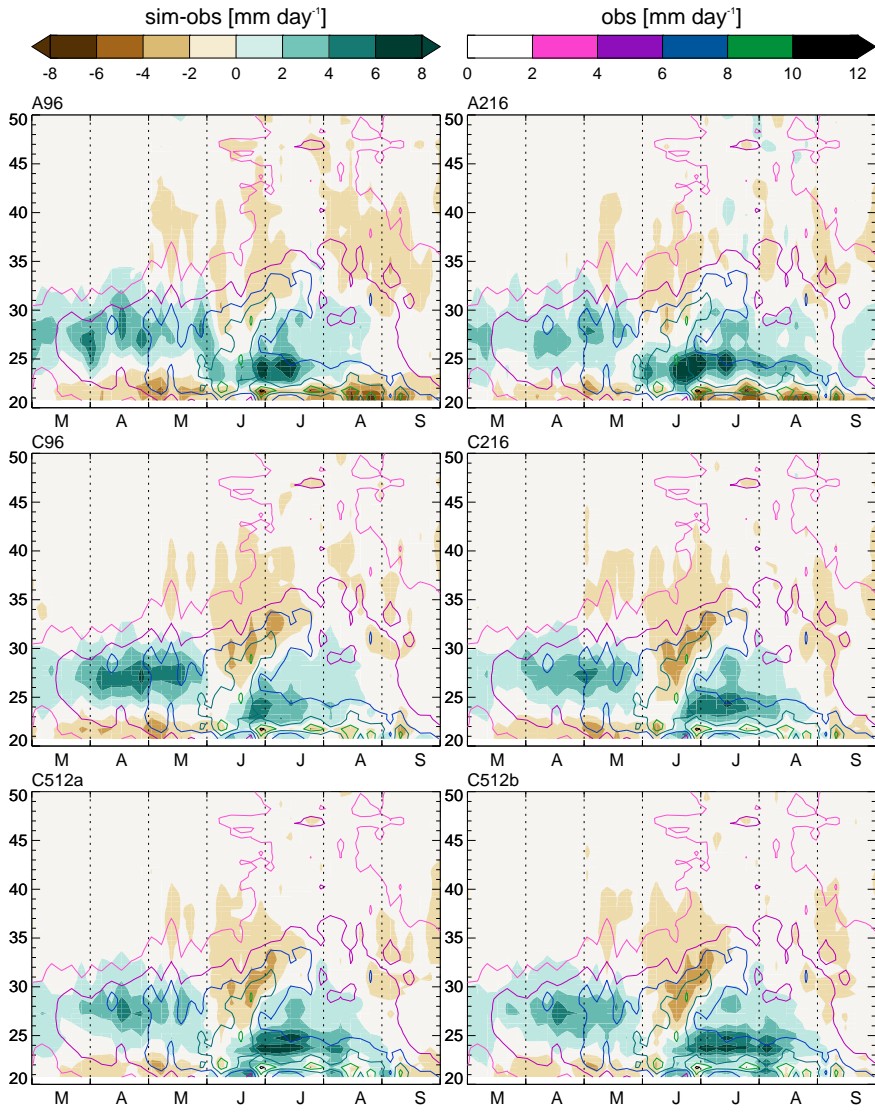

**Figure 3.** Hovmöller diagram showing the difference between simulated and observed May–September precipitation (shading), and observed precipitation (contours) averaged over 105–120° E as a function of pentad (x-axis) and latitude (y-axis), using the period 1982–2007 for observations and the full length of each simulation. All data are interpolated to the APHRODITE grid.





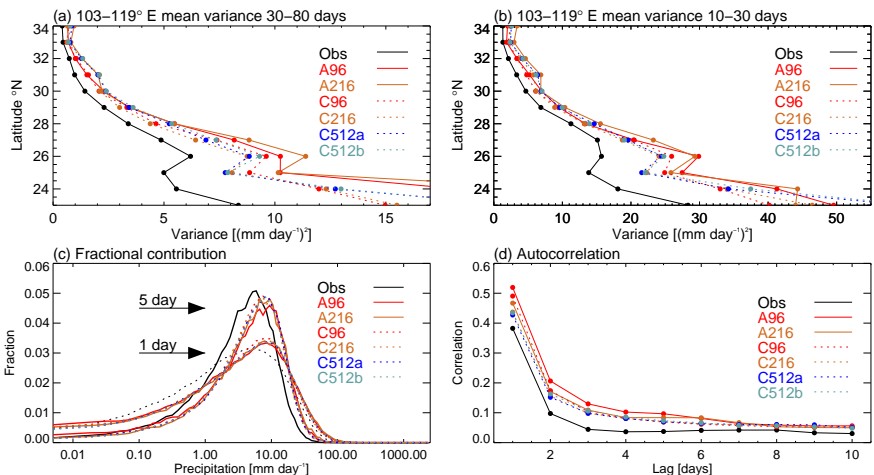

**Figure 4.** (a,b) Observed and simulated meridional profiles of the intraseasonal variance of precipitation averaged between 103–119° E for (a) 10–30-day and (b) 30–80-day filtered variability. (c,d) For precipitation averaged over southern China (22–32° N, 103–119° E) the (c) observed and simulated spectra of fractional contributions to daily and 5-day precipitation totals and (d) autocorrelations of daily precipitation at different time lags averaged over southern China (22–32° N, 103–119° E). Details on the computations are given in Sect. 2.5





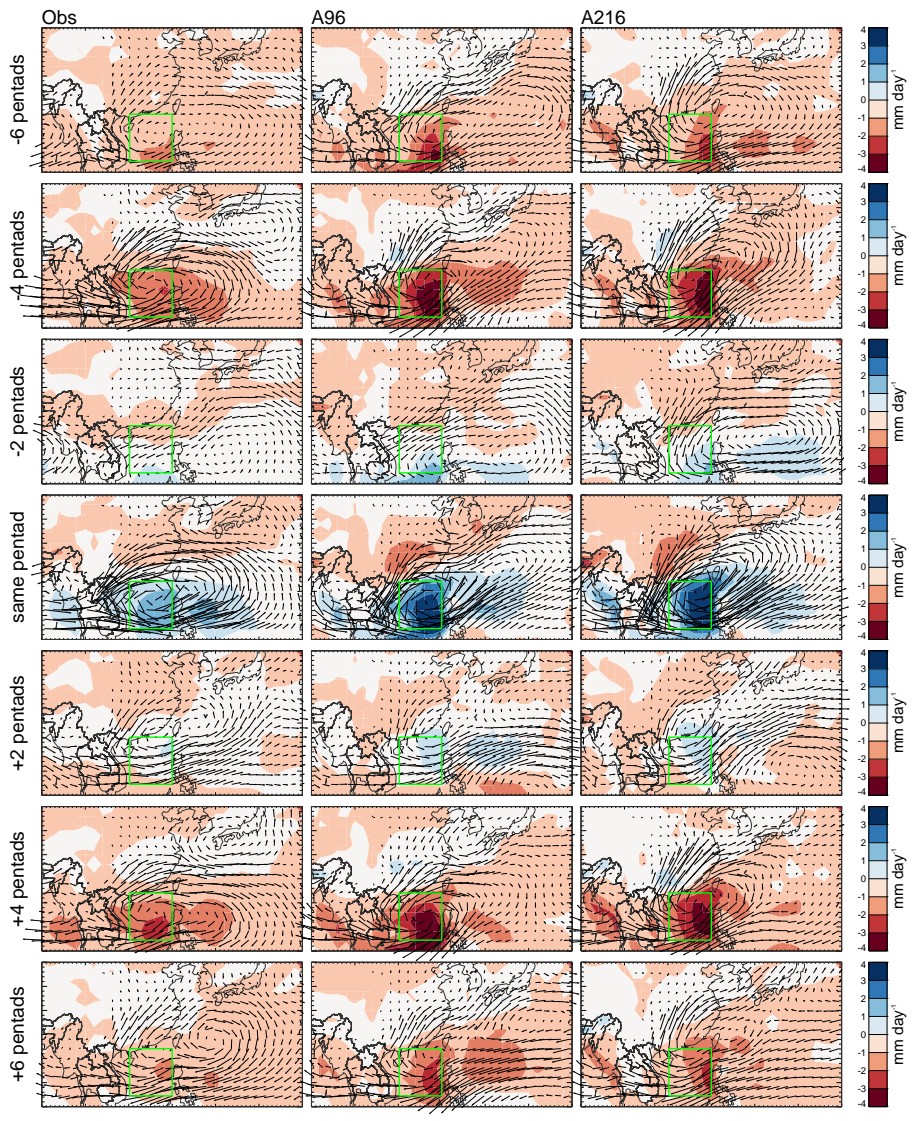

**Figure 5.** The regressed space-time evolutions of observed and simulated 30–80-day filtered precipitation and 850 hPa wind with respect to the normalized pentad rainfall timeseries averaged in the South China Sea (green box, 12–22° N, 110–120° E). Wind arrows of 1 ms$^{-1}$ correspond 5° in latitude or longitude.



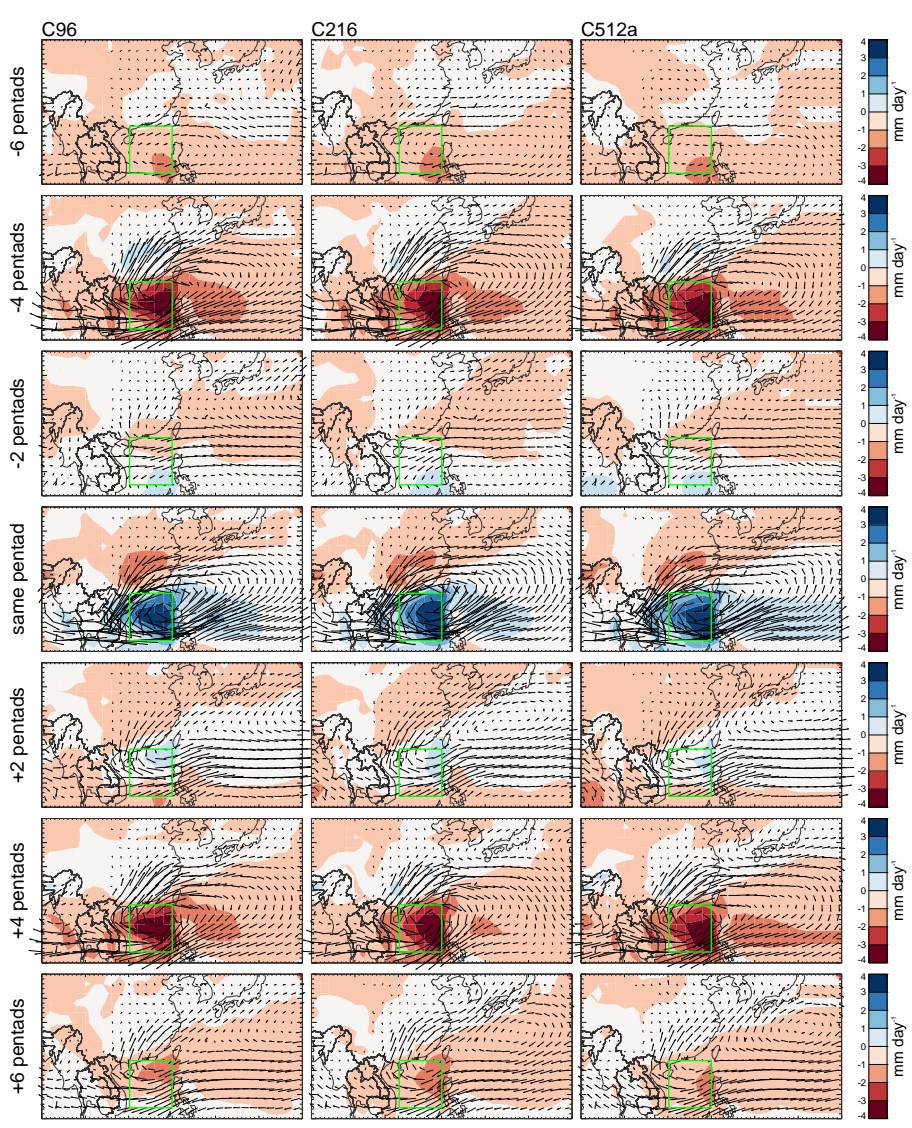

**Figure 6.** As in Fig. 5, but for C96, C216 and C512a.



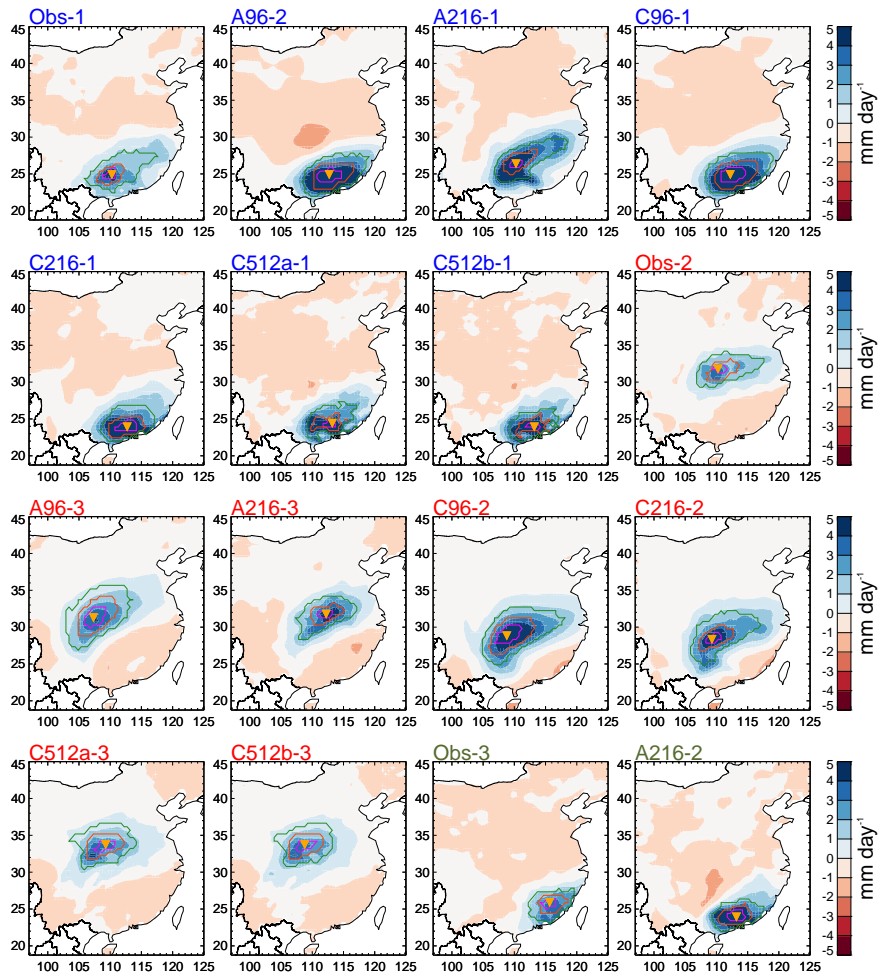

**Figure 7.** Observed (Obs) and simulated EOT patterns corresponding to Obs-1 (blue labels), Obs-2 (red labels) and Obs-3 (gray labels). Shading shows regressions of MJJASO precipitation against the normalized EOT timeseries. Contours show correlations of the full (leading order) or residual (higher order) precipitation-anomaly timeseries with the EOT base point exceeding 0.8 (magenta), 0.6 (orange), 0.4 (green). The EOT base point is marked by the orange inverted triangle.





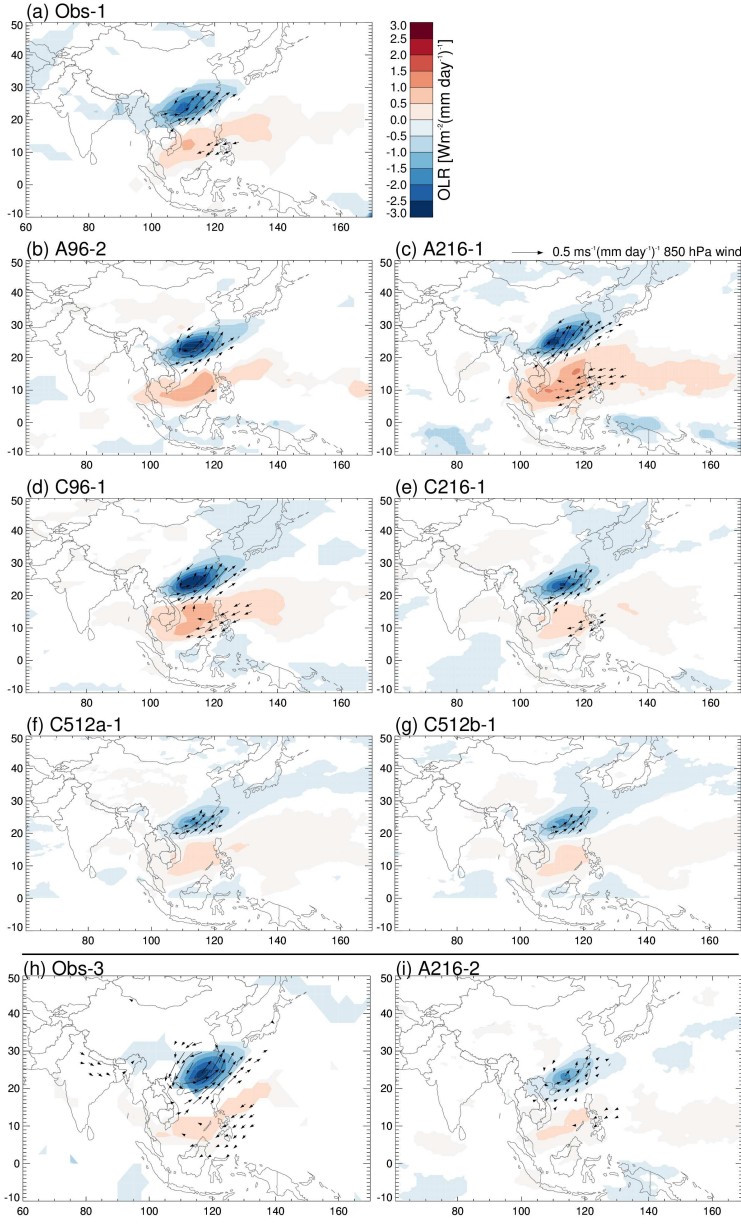

**Figure 8.** Regressions of OLR (shading) and 850 hPa wind (arrows) against MJJASO timeseries associated with Obs-1 (a-g) and Obs-3 (h-i). All shown values are significant at the 10 % level. Wind arrows are drawn where one component is statistically significant and the speed exceeds 0.1 ms$^{-1}$ (mm day$^{-1}$)$^{-1}$ (a-g) or 0.04 ms$^{-1}$ (mm day$^{-1}$)$^{-1}$ (h-i).



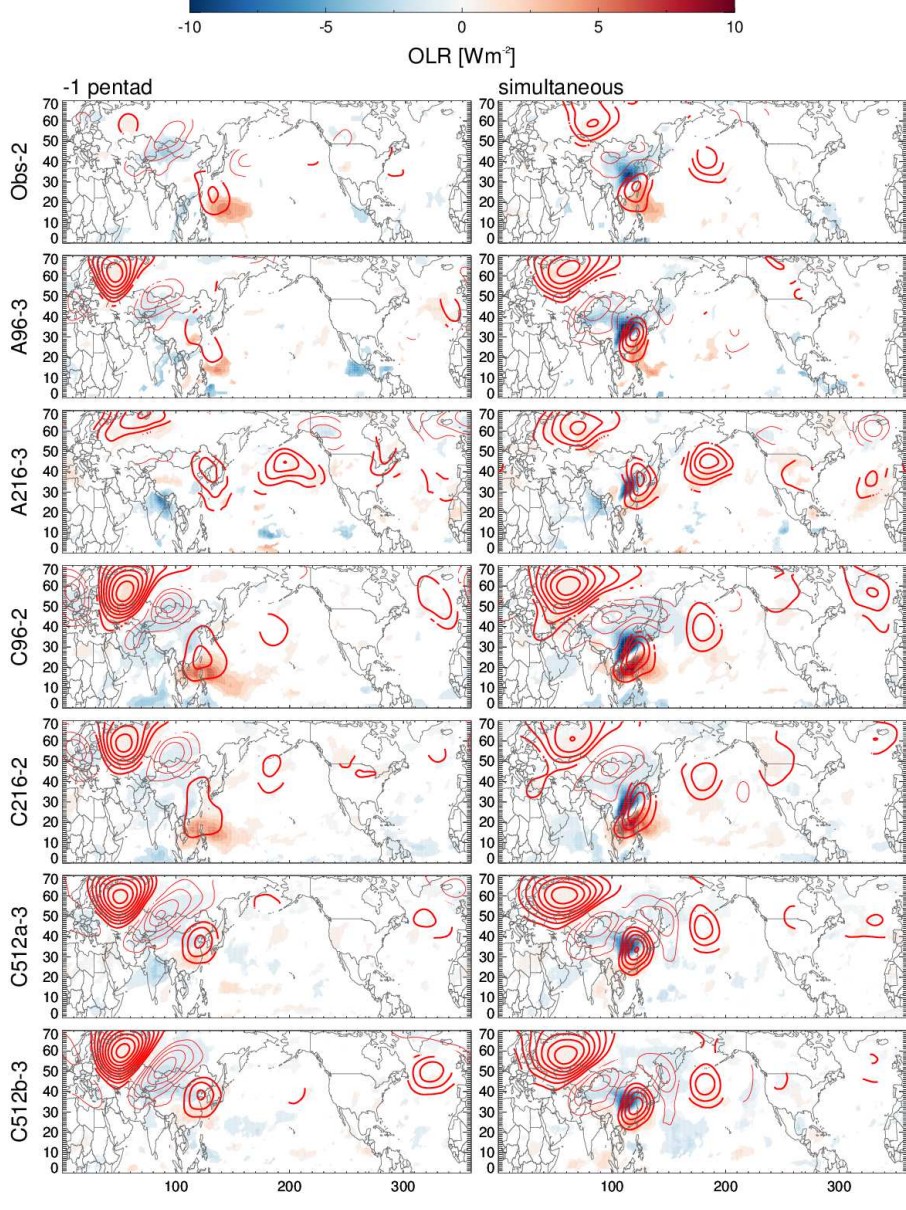

**Figure 9.** Regression maps of OLR (shading) and $Z_{500}$ (contours, thick: positive, thin: negative, intervals of 0.3 m (mm day$^{-1}$)$^{-1}$) against MJJASO EOT timeseries associated with Obs-2 at lead times of one (left), and zero (simultaneous) pentads. All values shown are significant at the 10 % level.



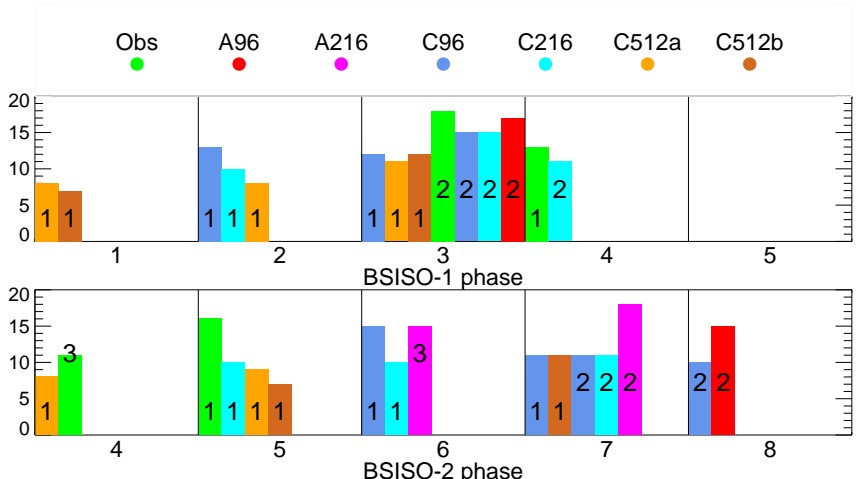

**Figure 10.** Histograms show the percentages of the wettest 10 % of pentads that occur during phases 1–5 of BSISO-1 (top) and phases 4–8 of BSISO-2 (bottom), when these are statistically significantly increased at the 5 % level compared to climatology, as indicated by a two-sided binomial test. Numbers overlaid on the histograms indicate the pattern number. BSISO phases that are not listed are not associated with any anomalously increased occurrences.

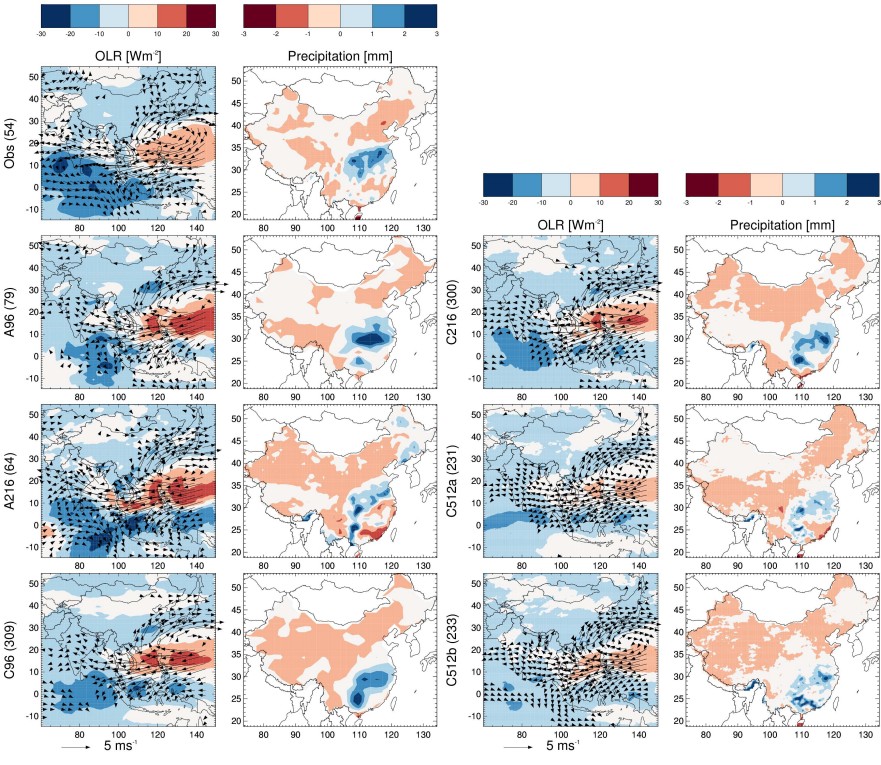

**Figure 11.** To illustrate simulation biases related to the BSISO, pentads with BSISO-1 phase 3 amplitudes ≥1 are chosen for composites of OLR, 850 hPa wind, and precipitation in units of mm day$^{-1}$. Values in brackets indicate the numbers of pentads that were used in each composite.





**Table 1.** The resolution, integration length, and type of ocean coupling are listed for all simulations. All simulations have 85 vertical levels with a model lid at 85 km.

| Simulation | Nodal number | Resolution at 50° N [km] | Integration length (years) | Coupling to ocean |
|---|---|---|---|---|
| A96 | N96 | 135 | 27 (1982–2008) | atmosphere-only |
| A216 | N216 | 60 | 27 (1982–2008) | atmosphere-only |
| C96 | N96 | 135 | 100 | coupled |
| C216 | N216 | 60 | 100 | coupled |
| C512a | N512 | 25 | 100 | coupled |
| C512b | N512 | 25 | 100 | coupled |





**Table 2.** Column (1) observed (Obs) and simulated (labeled by simulation name) EOT patterns in MJJASO; numbers indicate the order of the EOT pattern. (2) linear pattern correlation coefficient of simulated and observed precipitation anomalies, (3) explained space-time variance of the EOT pattern, (4) standard deviation of the EOT timeseries.

|  | Pattern corr. | Expl.Var.[%] | Stddev.[mm day$^{-1}$] |
|---|---|---|---|
| Obs-1 |  | 7 | 6 |
| A96-2 | 0.82 | 12 | 8 |
| A216-1 | 0.88 | 9 | 7 |
| C96-1 | 0.89 | 11 | 7 |
| C216-1 | 0.85 | 8 | 10 |
| C512a-1 | 0.83 | 7 | 12 |
| C512b-1 | 0.74 | 7 | 12 |
| Obs-2 |  | 4 | 4 |
| A96-3 | 0.77 | 5 | 4 |
| A216-3 | 0.90 | 4 | 5 |
| C96-2 | 0.70 | 10 | 6 |
| C216-2 | 0.62 | 7 | 6 |
| C512a-3 | 0.66 | 3 | 4 |
| C512b-3 | 0.68 | 3 | 4 |
| Obs-3 |  | 4 | 4 |
| A216-2 | 0.75 | 8 | 14 |