# Peer review of "Intraseasonal summer rainfall variability over China in the MetUM GA6 and GC2 configurations"

_Geoscientific Model Development, 2018_

## Referee Comment (RC1) · Anonymous Referee #1 · 25 Apr 2018

The authors have produced a nicely written article on the important topic of the performance of one of the MetUM models in simulating boreal summer intraseasonal variability over China. It is useful to quantify how well a model can reproduce such variability given its important impacts on water resources and extreme events during the monsoon season; yet it remains a very poorly understood phenomenon. The introduction is well motivated and references the appropriate literature; the techniques used are varied and sound, having been used comprehensively in this field before. I have several major comments on this paper which are as follows:

(1) In section 3, the authors give an assessment with a discussion of mean state biases.

Since the western Pacific subtropical high is another dominate climate system in East Asian region, evaluation on the simulation of the western Pacific subtropical high will gain more insight into the model bias in precipitation.

(2) In Section 4, the authors shows the meridional profiles of bandpass-filtered precipitation variance averaged between 103–119 E. The role of bias in 30–80 and 10–30 day variability on the total bias in IPV is explored. Why the authors divide the ISO over China into 10-30 days and 30-80 days? If you take an analysis on the power spectra of precipitation or OLR over the Eastern China, you can find that the biweekly oscillation (peak on 10-20days) is significant, ISO on time scale of 20-60 days (with one or two peaks) is also evident. Thus, in this section, investigating what timescales are associated with IPV biases according to the dominate period over China may be more meaningful. (3) In Section 5, the fractions of the wettest decile of pentads that occur during phases 1–5 of BSISO-1 and phases 4–8 of BSISO-2 are shown. Here, the BSISO-1 and BSISO-2 are usually used to represent the BSISO over the Asian region which including the Indian region. Previous studies have found that there are the large regional differences between the Indian region and WNP region. The former is characterized by northeastward propagation with period on 20–50 days, while the later show periodicity on 10-60 days, which includes a significant biweekly component. So if the analysis based on the BSISO index derived from the regional ISO over the East Asian – WNP region may be more convincing, and the discussion on the relationship between the BSISO phase and EOT mode may be more interesting.

(4) In Section 4, Fig.10 gives the percentages of the wettest 10 % of pentads that occur in different BSISO phase. Then, how about the occurrences of the percentages of the wettest 10 % of pentads in different month?
* * *

---

## Referee Comment (RC2) · Anonymous Referee #2 · 2 May 2018

In this manuscript, the authors investigate the role of coupling, resolution, and decadal variability on the simulation of intraseasonal summer variability over China using the MetUM climate model. This is a nice contribution to the climate modeling and prediction community efforts in understanding the relative importance of these factors for reducing the biases of climate models and improving the model prediction skill.

I have only major comment that the authors should address. Section 5: The higher correlation value of the EOT patterns in the uncoupled simulations suggests that these patterns might be driven by the SST rather than the atmospheric variability. The authors should consider evaluating the SST biases of the coupled simulations. Understanding

the role of coupling is not trivial because coupled models have biases that can interfere with the air-sea interaction processes.

Minor comments: Introduction: L 25 model formulation needs to be clarified Section 2.1: L25 naming convection-> naming convention Figure 4c does not show fractional contributions to daily precipitation totals in observations. Section 4: The slope of the autocorrelation curves is consistent between the model and observations. Define what is meant by the decorrelation time.

Section 5: At pentad -1 a negative Z500 anomaly is located over the northwest Pacific.

Discussion: The authors speculate that parameterization of convection can be another cause of the model biases. There are a few studies showing the impact of cloud processes parameterization of the simulation of summer rainfall variability over China (e.g., Chen et al. 2010, DeMott et al. 2013, Jin and Stan 2016).

Chen, H., T. Zhou, R. B. Neale, X. Wu, and G. Zhang, 2010: Performance of the New NCAR CAM3.5 in East Asian Summer Monsoon Simulations: Sensitivity to Modifications of the Convection Scheme, J. Climate, 23, 3657-3675, doi:10.1175/2010JCLI3022.1.

DeMott, C. A., C. Stan, and D. A. Randall, 2013: Northward propagation mechanisms of the boreal summer intraseasonal oscillation in the ERA-Interim and SP-CCSM. J. Climate, 26, 1973-1992, doi:10.1175/JCLI-D-12-00191.1

Jin, Y., and C. Stan, 2016: Simulation of East Asian Summer Monsoon (EASM) in SP-CCSM4. Part I: Seasonal mean state and intraseasonal variability. J. Geophys. Res., 121, 7801-7818, doi:10.1002/2015JD024035.
* * *

---

## Author Comment (AC1) · 26 Jun 2018

We thank the two anonymous referees for their helpful reviews. We addressed all comments as detailed below in italic font. Line numbers refer to the 'tracked-changes' document.

**Referee 1**

The authors have produced a nicely written article on the important topic of the performance

of one of the MetUM models in simulating boreal summer intraseasonal variability

over China. It is useful to quantify how well a model can reproduce such variability

given its important impacts on water resources and extreme events during the monsoon

season; yet it remains a very poorly understood phenomenon. The introduction

is well motivated and references the appropriate literature; the techniques used are varied

and sound, having been used comprehensively in this field before. I have several

major comments on this paper which are as follows:

(1) In section 3, the authors give an assessment with a discussion of mean state biases.

Since the western Pacific subtropical high is another dominate climate system in East

Asian region, evaluation on the simulation of the western Pacific subtropical high will

gain more insight into the model bias in precipitation.

*We agree that an evaluation of the simulation of the WNPSH should be added. We added (now) Figure 4. It shows the strength and location of the WNPSH in observations and the simulations in June, and the ridge position in June, July and August. This figure is now discussed at lines P6 L16-20 where it is connected to the model bias in precipitation: 'This lack of northward progression in the simulations may be linked to a poor simulation of the western North Pacific subtropical high (WNPSH; Fig. 4). In GA6 the WNPSH is too weak; in C96 and C216 it is too weak and does not extend far enough westward; in C512a and C512b the strength of the WNPSH is closer to observations to the east of 130◦ E, but the western flank remains too weak. The northward propagation of the WNPSH ridge in 110◦ E–120◦ E from June to August is not correctly captured in any simulation.'*

(2) In Section 4, the authors shows the meridional profiles of bandpass-filtered precipitation

variance averaged between 103–119 E. The role of bias in 30–80 and 10–30

day variability on the total bias in IPV is explored. Why the authors divide the ISO

over China into 10-30 days and 30-80 days? If you take an analysis on the power

spectra of precipitation or OLR over the Eastern China, you can find that the biweekly

oscillation (peak on 10-20days) is significant, ISO on time scale of 20-60 days (with

one or two peaks) is also evident. Thus, in this section, investigating what timescales

are associated with IPV biases according to the dominate period over China may be

more meaningful.

*This is a valid point. We modified the filtering windows to 10-20 and 20-60 day. Figure (now) 5 and the text at lines P6 L27-28 has been updated accordingly. This change did not affect the conclusions.*

(3) In Section 5, the fractions of the wettest decile of pentads that

occur during phases 1–5 of BSISO-1 and phases 4–8 of BSISO-2 are shown. Here,

the BSISO-1 and BSISO-2 are usually used to represent the BSISO over the Asian region

which including the Indian region. Previous studies have found that there are the

large regional differences between the Indian region and WNP region. The former is

characterized by northeastward propagation with period on 20–50 days, while the later

show periodicity on 10-60 days, which includes a significant biweekly component. So

if the analysis based on the BSISO index derived from the regional ISO over the East

Asian – WNP region may be more convincing, and the discussion on the relationship

between the BSISO phase and EOT mode may be more interesting.

*In principle, this may be correct. However, we are specifically interested in the effects of the large-scale BSISO on East Asia, rather than the effects of the regional manifestation of the BSISO over the western North Pacific, which may or may not be connected to the broader phenomenon. Previous studies have also used the Lee et al. (2013) BSISO indices for East Asia, e.g.:*

*https://journals.ametsoc.org/doi/full/10.1175/JCLI-D-16-0505.1*

*https://journals.ametsoc.org/doi/full/10.1175/JCLI-D-16-0206.1*

*https://rmets.onlinelibrary.wiley.com/doi/abs/10.1002/joc.4433*

*We modified the sentence at line P7 L1 to state our intention explicitly.*

*Also, our previous work with observations showed a connection between the large-scale BSISO and regional precipitation over East Asia, We therefore chose to continue to use the large-scale BSISO when evaluating the models, for consistency with our previous analysis of observations.*

(4) In Section 4, Fig.10 gives the percentages of the wettest 10 % of pentads that occur

in different BSISO phase. Then, how about the occurrences of the percentages of the

wettest 10 % of pentads in different month?

*We added Table 3 which shows the same information as Fig. 10, but split by month. It shows that the agreement between observations and the simulations is poor throughout the extended summer season. Fig. 10 is discussed at lines P8 L27-29.*

**Referee 2**

In this manuscript, the authors investigate the role of coupling, resolution, and decadal variability on the simulation of intraseasonal summer variability over China using the MetUM climate model. This is a nice contribution to the climate modeling and prediction community efforts in understanding the relative importance of these factors for reducing the biases of climate models and improving the model prediction skill.

I have only major comment that the authors should address. Section 5: The higher correlation value of the EOT patterns in the uncoupled simulations suggests that these patterns might be driven by the SST rather than the atmospheric variability. The authors should consider evaluating the SST biases of the coupled simulations. Understanding the role of coupling is not trivial because coupled models have biases that can interfere with the air-sea interaction processes.

*We agree that the manuscript was lacking a discussion of the issue of coupled model SST biases. We have addressed this point by adding a paragraph to the Discussion at lines P9 L10-16: 'However, it is difficult to isolate the effects of air-sea coupling, as mean state circulation biases are present in both GA6 and GC2. GC2 shows cold SST biases in the northern midlatitude Pacific and North Atlantic and warm biases over the Southern Ocean (Fig. 3 in Stephan et al., 2017c). These SST biases may be partly driven by atmospheric circulation biases and partly by errors in oceanic processes (dynamics and mixing). SST biases in coupled GCMs can strongly influence tropical convective variability (e.g., Klingaman and Woolnough, 2014; DeMott et al., 2015). However, it is not possible to isolate the effects of SST biases on rainfall patterns without performing separate simulations that correct the ocean state, for example, by imposing flux corrections, which are outside the scope of this study.'*

Minor comments:

Introduction: L 25 model formulation needs to be clarified

*The sentence is actually redundant and has been deleted.*

Section 2.1: L25 naming convection-> naming convention

*This typo has been corrected.*

Figure 4c does not show fractional contributions to daily precipitation totals in observations.

*We are grateful that this mistake was spotted. The line for observations has been added to (now) Fig. 5.*

Section 4: The slope of the autocorrelation curves is consistent between the model and observations. Define what is meant by the decorrelation time.

*We agree that we used the word ' decorrelation time' inappropriately. The sentence has been rephrased at line P6 L32.*

Section 5: At pentad -1 a negative Z500 anomaly is located over the northwest Pacific.

*This may have been ambiguous. We replaced 'northwest Pacific with 'East Asia and the western Pacific', line P8 L10.*

Discussion: The authors speculate that parameterization of convection can be another

cause of the model biases. There are a few studies showing the impact of cloud processes

parameterization of the simulation of summer rainfall variability over China (e.g.,

Chen et al. 2010, DeMott et al. 2013, Jin and Stan 2016).

*We thank the reviewer for these references and are now citing them at line P10 L9-10.*

Chen, H., T. Zhou, R. B. Neale, X. Wu, and G. Zhang, 2010: Performance

of the New NCAR CAM3.5 in East Asian Summer Monsoon Simulations: Sensitivity

to Modifications of the Convection Scheme, J. Climate, 23, 3657-3675,

doi:10.1175/2010JCLI3022.1.

DeMott, C. A., C. Stan, and D. A. Randall, 2013: Northward propagation mechanisms

of the boreal summer intraseasonal oscillation in the ERA-Interim and SP-CCSM. J.

Climate, 26, 1973-1992, doi:10.1175/JCLI-D-12-00191.1

Jin, Y., and C. Stan, 2016: Simulation of East Asian Summer Monsoon (EASM) in SPCCSM4.

Part I: Seasonal mean state and intraseasonal variability. J. Geophys. Res.,

121, 7801-7818, doi:10.1002/2015JD024035.

---

## Author Response (AR2)

Reviewer's comment:
It would be better if the authors give the regressed space–time evolution of precipitation and 850 hPa wind with periods of 20–70 days instead of 30-80 days in Fig. 6 and 7.

We modified the filtering window accordingly and replaced Figs 6 and 7 with plots based on 20–70 day bandpassed data. '30-80' was replaced with '20-70' in the text. No other changes were required as the modification did not affect any results.